ecology

mortality, community shift, disturbance, climate change, benthic communities, novel ecosystems

# Modelling the linkage between coral assemblage structure and pattern of environmental forcing

Stuart A. Sandin[1], Yoan Eynaud[1], Gareth J. Williams[1,2], Clinton B. Edwards[1] and Dylan E. McNamara[3]

[1]Scripps Institution of Oceanography, UC San Diego, 9500 Gilman Drive, La Jolla, CA 92093-0202, USA
[2]School of Ocean Sciences, Bangor University, Anglesey LL59 5AB, UK
[3]Department of Physics and Physical Oceanography/Center for Marine Science, University of North Carolina, Wilmington, 601 South College Road, Wilmington, NC 28403, USA

SAS, 0000-0003-1714-4492; GJW, 0000-0001-7837-1619;
CBE, 0000-0003-4222-0290; DEM, 0000-0001-8752-1586

Geographical comparisons suggest that coral reef communities can vary as a function of their environmental context, differing not just in terms of total coral cover but also in terms of relative abundance (or coverage) of coral taxa. While much work has considered how shifts in benthic reef dynamics can shift dominance of stony corals relative to algal and other benthic competitors, the relative performance of coral types under differing patterns of environmental disturbance has received less attention. We construct an empirically-grounded numerical model to simulate coral assemblage dynamics under a spectrum of disturbance regimes, contrasting hydrodynamic disturbances (which cause morphology-specific, whole-colony mortality) with disturbances that cause mortality independently of colony morphology. We demonstrate that the relative representation of morphological types within a coral assemblage shows limited connection to the intensity, and essentially no connection to the frequency, of hydrodynamic disturbances. Morphological types of corals that are more vulnerable to mortality owing to hydrodynamic disturbance tend to grow faster, with rates sufficiently high to recover benthic coverage during inter-disturbance intervals. By contrast, we show that factors causing mortality without linkage to morphology, including those that cause only partial colony loss, more dramatically shift coral assemblage structure, disproportionately favouring fast-growing tabular morphologies. Furthermore, when intensity and likelihood of such disturbances increases, assemblages do not adapt smoothly and instead reveal a heightened level of temporal variance, beyond which reefs demonstrate drastically

**Author for correspondence:**
Stuart A. Sandin
e-mail: ssandin@ucsd.edu

reduced coral coverage. Our findings highlight that adaptation of coral reef benthic assemblages depends on the nature of disturbances, with hydrodynamic disturbances having little to no effect on the capacity of reef coral communities to resist and recover with sustained coral dominance.

## 1. Introduction

Coral reef communities are extremely variable in both space and time, reflective of the interaction of species' biological traits with myriad environmental and anthropogenic drivers across scales [1–3]. Many of these drivers are predicted to change in frequency and intensity at unprecedented rates in the coming decades [4–6]. Shifts of coral ecosystem states are likely to follow [7], and in some cases the reef's plasticity will not be sufficient to maintain the recent-historical community state and the reef may transition to an alternate regime [8,9].

Long-term observations of coral reef benthic community composition offer insights into potential patterns of change under shifting environmental conditions. A 30 year investigation of one well-studied region of the Great Barrier Reef linked the types and scales of disturbance in coral-reef habitats to the structure of the reef benthic community. This work revealed a spectrum of realized community transitions, including limited change, change of coral taxonomic composition, and, in more dramatic cases, competitive shifts of benthic groups from corals to algae or non-coral invertebrates [10]. Heuristic models can describe patterns of benthic shift as a dynamic between the impact of the disturbance and the taxon-specific rate of regrowth [11]. Using palaeoecological approaches, transitions among coral taxa have been documented in multiple geographies as responses to shifting patterns of environmental disturbance (more recently linked to human activities) [12–20]. Recent evidence based upon *in situ* survey data is emerging, suggesting that some reefs may be able to reorganize following disturbance, maintaining a coral-dominated state but with a novel taxonomic composition [21,22]. However, spatially expansive survey data are only beginning to emerge offering robust views of stability or change in long-term benthic state (described as an 'attractor' of benthic condition, using terminology from dynamical systems), and how these benthic states vary across environmental conditions [23–25]. Numerical modelling provides an important tool for setting expectations and articulating hypotheses regarding how the behaviour of benthic dynamical systems may be linked to spatial variability of conditions and to change in these conditions over decadal scales.

A key dynamic leading to plasticity within a reef benthic assemblage is the set of characters and trade-offs implicit among life histories of constituent species. Corals, for example, are a modular taxon that can suffer whole or partial mortality in response to pulse disturbance events, such as tropical cyclones, ocean-warming events that trigger coral bleaching, disease and coral-predation. Extreme hydrodynamic disturbances (e.g. storms and cyclones) often can cause whole colony mortality [26,27], particularly in fast-growing coral taxa with large surface areas that are vulnerable to dislodgement [28]. By contrast, disturbances that can induce partial mortality (e.g. coral bleaching, disease and predation) appear to have a less consistent and comparable bias towards fast- or slow-growing coral taxa [29–32]. As such, the taxonomic 'winners' and 'losers' often differ depending on the disturbance in question and the predominant mortality type that follows.

As the frequency and intensity of contrasting disturbances and subsequent mortality events change in the future, what will be the resulting impact to coral-reef benthic communities, and which scenarios will favour shifts to novel coral assemblages? To answer this, we use an empirically-grounded numerical model to explore the effects of the frequency and intensity of two contrasting mortality types on coral assemblage structure: (i) whole colony mortality owing to mechanical dislodgment following storms or other major hydrodynamic events; and (ii) partial to whole colony mortality following coral bleaching, disease, coral predation or other colony-specific disturbance events. The model is parametrized to explore differing patterns of survival of coral morphological types, without extending into scenarios leading to systematic transitions resulting in functional extirpation of hard corals (e.g. removal of herbivores resulting in replacement of corals by algal competitors [33,34]).

## 2. Material and methods

Coral demographic rates, their response to disturbance and their competitive dynamics are linked strongly to colony morphology [35]. The complexity of coral reef ecosystems and their quality as a

**Table 1.** Model parameters, including explicit value (for biological parameters) and ranges (for disturbance parameters), along with definition, units and source.

| symbol | value | interpretation | units | source |
|---|---|---|---|---|
| $\alpha_D$ | 1.25 | *digitate* coral baseline sensitivity | — | calculated from [28] |
| $\alpha_M$ | −0.94 | *massive* coral baseline sensitivity | — | calculated from [28] |
| $\alpha_T$ | 4.47 | *tabular* coral baseline sensitivity | — | calculated from [28] |
| $\beta_D$ | −0.036 | *digitate* coral sensitivity term | — | calculated from [28] |
| $\beta_M$ | −0.233 | *massive* coral sensitivity term | — | calculated from [28] |
| $\beta_T$ | 0.385 | *tabular* coral sensitivity term | — | calculated from [28] |
| $C_R$ | 0.01 | size of a recruit | $m^2$ | [33] |
| $DMT_H$ | 10–250 | intensity of the hydrodynamic disturbance | — | [27] |
| $DMT_N$ | 500 | intensity of the hydrodynamic regime baseline | — | — |
| $g_D$ | 22 | digitate coral annual growth rate (*Acropora humilis*) | mm | [38,39] |
| $g_M$ | 7 | massive coral annual growth rate (*Goniastrea retiformis*) | mm | [40,41] |
| $g_T$ | 55 | tabular coral annual growth rate (*Acropora cytherea* and *Acropora hyacinthus*) | mm | [38,39,42] |
| $\kappa_0$ | 0.01–0.5 | size of the area lost by a colony impacted by colony-specific disturbance event | $m^2$ | — |
| $\rho_H$ | 0–1 | annual probability of a hydrodynamic disturbance | — | — |
| $\rho_R$ | 25 | annual recruitment rate | $m^{-2}$ | [33] |
| $r_0$ | 0.01–0.95 | ratio of the population impacted by the colony-specific disturbance event | — | — |
| $\rho_0$ | 0–1 | annual probability of a colony-specific disturbance event | — | — |
| $S_Q$ | 25 | size of the studied area available to coral | $m^2$ | — |

habitat for other reef-associated organisms depends greatly on the abundance of morphological types [36]. Here we characterize corals as belonging to one of three morphological groups: (i) tabular/foliose; (ii) digitate/corymbose/sub-massive, and (iii) encrusting/massive. Note that we construct this model using major morphological groups typical in forereef habitats of Pacific coral reefs between 8–20 m depth [2,25,37].

Life-history strategies and demographic rates can be generalized for coral species typical of each morphological functional group. Growth rates differ greatly across morphologies, with tabular/foliose corals growing on average faster than digitate/corymbose/sub-massive, themselves growing faster than encrusting/massive (table 1). Further, Madin and colleagues have established a quantitative link between the morphology and size (planar area) of a coral colony and its dislodgement probability following a given hydrodynamic event [27,28]. We use data previously published [28] to parametrize the equation linking the colony size of a given morphology to its specific vulnerability to a hydrodynamic disturbance event. We represent hydrodynamic disturbance as any anomalous event (i.e. higher than the baseline intensity), ranging from a large wave event (low intensity) to extremely high sea states generated by large tropical cyclones (high intensity). In the model, the sensitivity to a hydrodynamic event of a given intensity tends to increase with colony size for tabular/foliose corals, is approximately constant with size for digitate/corymbose/sub-massive corals and decreases with size for encrusting/massive corals (see table 1 for parameter ranges and literature sources).

Unlike hydrodynamic disturbances, there is little evidence for consistent mechanistic links between a coral's morphology and its quantitative impacts from colony-specific disturbances. We use the term 'colony-specific disturbance' here to refer to the collection of physical and biological disturbances that have individualized impacts on a coral colony (e.g. disease, environmental stress leading to bleaching, predation). Such disturbances are operationally distinct from hydrodynamic disturbance in two

principal ways. First, colony-specific disturbances introduce a probabilistic mortality for all colonies independent of morphology, while hydrodynamic disturbances introduce a physically determined mortality (defined by deterministic physical dislodgement criteria under a particular level of wave energy) that is morphology-dependent and results in whole-colony mortality. Second, colony-specific mortality can lead to a range of potential impacts on coral colonies, from partial mortality through to whole-colony loss. Importantly, such forms of mortality often have size-structured impacts, in that smaller colonies are more likely to suffer whole colony mortality from such disturbances. Further, this collection of colony-specific disturbances, while often demonstrating taxonomic specificity [32,43], reveals limited systematic morphological bias. Hence, here we follow a simplified approach to describe patterns of mortality linked to colony-specific disturbances; when a given disturbance event occurs, it only impacts a proportion of the coral colony population, leading to the loss of a given area for each of the impacted colonies (a size-specific bias in proportion of colony lost), independent of morphology.

## 2.1. The model

The model uses the following recursive equations (with 1-year time steps) to simulate the net growth of individual coral colonies in two-dimensional space:

$$C_{i,j,t} = \begin{cases} C_{i,j,t-1} + G_{i,j,t} - \mu_{i,j,t} & \text{if } X_{i,j,t} = 0 \\ 0 & \text{if } X_{i,j,t} = 1, \end{cases} \tag{2.1}$$

where $C_{i,j,t}$ is coral planar area, $G_{i,j,t}$ is area added owing to growth, $\mu_{i,j,t}$ is the function for tissue loss from colony-specific mortality and $X_{i,j,t}$ is a binary function for loss from hydrodynamic disturbance, each describing the $i$th colony, of type $j$, at time $t$. We assume that the colonies grow circularly, and thus the growth function $G$ is defined as:

$$G_{i,j,t} = \left(2\pi g_j \sqrt{\frac{C_{i,j,t-1}}{\pi}} + \pi g_j^2\right) \frac{\left(S_Q - \sum_{j=1}^{3} \sum_{i=1}^{n_{j,t}} C_{i,j,t-1}\right)}{S_Q}, \tag{2.2}$$

with $G_{i,j,t}$ being the planar area growth of colony $i$ of type $j$ at that time $t$, and $g_j$ the absolute radial growth of a given type $j$. Note that the assumption of constant radial growth rate, independent of size, is consistent with models of allometric growth of multiple coral taxa [44,45]. $S_Q$ is the total area of substrate available to coral (table 1) and the summation is the total area covered by the three morphological types (arbitrarily coded—(i) tabular/foliose, (ii) digitate/corymbose/sub-massive and (iii) encrusting/massive) across all $n_{j,t}$ colonies per type $j$ at time $t$; the ratio in the second term serves to scale growth proportionally to the area available to new coral growth. This assumption of growth being proportional to amount of free space ignores specifics of spatially explicit competition, especially under conditions of extreme clustering [46]. An exploration of spatially explicit models parametrized similarly as here have shown qualitatively similar behaviour for a range of parameter values included in this exercise, as extreme clustering is rarely encountered without forcing models through defined starting conditions of introduction of heterogeneous spatial landscapes (results not presented here).

In the model, hydrodynamic disturbances and colony-specific disturbances occur with probabilities of $\rho_H$ and $\rho_o$, respectively. The primary distinction between the two forms of disturbance is linked to the scale of impact—hydrodynamic disturbances are physical forcings that uniformly affect coral colonies at a site, while colony-specific disturbances are the collection of focused impacts that affect individual colonies independently across a site. Further, hydrodynamic disturbances cause only whole colony mortality, while colony-specific disturbances can cause a range of impacts, from partial mortality through to whole colony mortality. We define colony-specific disturbances here to be those that are above and beyond baseline life-and-death processes (which are embedded here within the growth function), and instead depict episodic events that have impacts which are manifested through tissue mortality of individual colonies (e.g. disease, thermal-induced bleaching, predation).

The probability that a hydrodynamic disturbance causes whole-colony mortality is a function of the intensity of the disturbance (e.g. wave power), strength of the substrate, the morphology of the coral, and the size of the coral. A colony will be dislodged if its *colony shape factor* value, $CSF_{i,j,t}$, is greater or equal to a given threshold linked to the disturbance intensity, the *dislodgment*

*mechanical threshold* (DMT) [27]. The value of $CSF_{i,j,t}$ for colony $i$, of morphological type $j$, at time $t$, is written as:

$$\ln(CSF_{i,j,t}) = \beta_j \ln(C_{i,j,t}) + \alpha_j + \varepsilon_j$$
$$\text{and } \varepsilon_j | C_{i,j,t} \tilde{E}_j, \quad (2.3)$$

where $\beta_j$ is the sensitivity scaling term for morphological type $j$, $\alpha_j$ is the baseline sensitivity term for morphological type $j$, and $\varepsilon_j$ is a random variable drawn from the empirically calculated residual distribution of morphological type $j$, $E_j$ (calculated from [28]). The probability, $X_{i,j,t}$ for colony $i$ of type $j$ at time $t$ to be dislodged by a hydrodynamic event of intensity DMT is defined as:

$$X_{i,j,t} = \begin{cases} 1, & CSF_{i,j,t} \geq DMT \\ 0, & CSF_{i,j,t} < DMT \end{cases} \quad (2.4)$$

and

$$DMT = \max\{DMT_H \, Z(1,\rho_H); \, DMT_N\},$$

where $Z$ is a draw from a binomial distribution with one trial and probability $\rho_H$, $DMT_H$ is the intensity of a hydrodynamic disturbance and $DMT_N$ the hydrodynamic disturbance baseline.

In this model, all episodic disturbances not caused by hydrodynamic disturbance impact colonies randomly across the coral population. Corals, like most colonial organisms, can exhibit both whole- and partial-colony mortality as a result of disturbances that affect the survivorship of tissue (e.g. necrosis or predation not leading to dislodgement of the entire colony). Such colony-specific disturbances will be described as the loss of a given area of live tissue from an individual coral. We define the mortality term which represents the loss of planar area owing to such episodic, local, colony-specific disturbances as $\mu_{i,j,t}$, the colony-specific areal loss for colony $i$ of morphological type $j$ at time $t$. When the binomial draw for time $t$ with probability $\mu_0$ reports no local disturbance, $\mu_{i,j,t} = 0$ for all colonies. When, instead, a local disturbance is drawn for the colonies in the modelled landscape, tissue loss is described as:

$$\mu_{i,j,t} = \begin{cases} 0 & \text{if } Y(1,r_o) = 0 \\ \kappa_0 & \text{if } Y(1,r_o) = 1 \text{ and } C^*_{i,j,t} > \kappa_0 \\ C^*_{i,j,t} & \text{if } Y(1,r_o) = 1 \text{ and } C^*_{i,j,t} \leq \kappa_0 \end{cases} \quad (2.5)$$

and

$$C^*_{i,j,t} = C_{i,j,t-1} + G_{i,j,t},$$

where $\kappa_0$ is the maximum area of tissue lost when a colony is affected by such disturbances and $Y$ is a draw from a binomial distribution with one trial and probability $r_0$. Equation (2.5) implicitly contains size-dependence of colony-specific disturbance, as smaller colonies will lose proportionately more tissue, or will suffer complete mortality more likely, than larger colonies. Note that $C^*_{i,j,t}$ is an operational placeholder, as growth step is considered prior to the step of partial mortality. If $C_{i,j,t} = 0$, the colony is removed from the system and not considered in subsequent time steps (i.e. the colony 'dies' in the model).

Colonies are only recruited into the modelled population at a given size, $C_R = 0.01 \text{ m}^2$, and following a Poisson distribution with a coefficient rate $\rho_R$ equal to 25 individuals $\text{m}^{-2} \text{yr}^{-1}$ (table 1). Note that we use the threshold of $0.01 \text{ m}^2$ within the model, as our target is to study dynamics of juvenile and adult populations, and smaller early-life-history individuals appear to follow distinct dynamics [47]. The results presented in this study are not sensitive to shifts in the defined size of a recruit (e.g. spanning definitions of coral recruits from 1–10 cm$^2$), as model outputs are units of summed coral area, the dynamics of which are dominated by growth patterns of larger adult corals (analyses not presented here).

## 2.2. Simulation model

In the model, mortality patterns are controlled by five parameters: (i) $\rho_0$, the probability of a colony-specific disturbance event; (ii) $r_0$, the expected proportion of colonies impacted by a colony-specific disturbance event; (iii) $\kappa_0$, the maximum tissue area lost per colony following a colony-specific disturbance; (iv) $\rho_H$, the probability of a hydrodynamic disturbance; and (v) $DMT_H$, the intensity of a hydrodynamic disturbance. For each simulation, parameters were selected from a range of values (table 1). Note that the ranges were defined to span values describing the wide variety of forcing conditions observed across reefs.

We characterized the coral assemblage by measuring the mean area covered through time, relative to the total area, by each morphological type (T, tabular/foliose; D, digitate/corymbose/sub-massive;

M, the encrusting/massive):

$$A = \left[ \sum_{t=1}^{t_{max}} \sum_{i=1}^{n_{M,t}} C_{i,M,t}; \; \sum_{t=1}^{t_{max}} \sum_{i=1}^{n_{D,t}} C_{i,D,t}; \; \sum_{t=1}^{t_{max}} \sum_{i=1}^{n_{T,t}} C_{i,T,t} \right] \Big/ t_{max}, \qquad (2.6)$$

where $n_{j,t}$ are the number of colonies of type $j$ at time $t$ and $t_{max}$ is the total number of time steps in the simulation.

We explored the impacts of interactions between hydrodynamic and colony-specific disturbance events by simulating the population dynamics over 500 years for a set of 100 000 different disturbance scenarios, with values ranging from 0 to 1 for $\rho_0$, 0.01 to 0.95 for $r_0$, from 0.01 to 0.5 m$^2$ for $\kappa_0$, from 0 to 1 for $\rho_H$ and from 250 to 10 for DMT$_H$ (low intensity to high intensity). Each simulation reached the attractor early in the simulation (typically within 30–70 years), and the window of 500 years was used to assure that a robust estimate of the steady state, or attractor, and the temporal variance around the attractor, could be reported. The final configuration was deemed an attractor in the sense that simulations starting from different initial conditions (not shown) evolved to a similar state. No evidence was found suggesting that any model had multiple attractors, and a complete analysis of the attractor landscape is beyond the scope of this contribution.

Models were initialized as empty landscapes with no coral. Coral recruitment and growth filled the landscape in the first years of simulation, with the time until reaching the attractor ranging from 20 to 100 years for most parameter combinations. Given the explicitly stochastic nature of the model (i.e. a simulation of stochastic disturbances), the benthic configuration was not static through time but instead varied around a mean benthic configuration. We use the mean and variance (see below) of benthic configuration over the entire simulation period to describe important elements of the attractor [46].

## 2.3. Output analysis

Our analysis aims to identify the independent and interacting effects of each parameter describing how disturbances affect the configuration of the resulting assemblage. Given the wide range of parameter combinations and system non-linearity, we chose to analyse the mean states resulting from simulation results across disturbance scenarios using regression tree analysis [48]. A multivariate regression tree was constructed through recursive partitioning of the variable space $\psi = (\rho_0, r_0, \kappa_0, \rho_H, DMT_H)$ based upon sets of outputs, $A$. The analysis defines subsets (classes) of the response variable space (the vector of mean benthic cover by group over 500 years of simulation) for which the values are the most similar. The similarity of each class is measured using a deviance criterion that has been normalized by the output size of the response variable. The output of the splitting procedure is presented as the parameter breaks defining each split. Further, the stereotypical assemblage for each terminal node is described as the multivariate mean of coral cover across all simulated assemblages defined by this node. See the electronic supplementary material, appendix S1 for more details of the splitting procedure used here.

While the regression tree analysis created classes based upon relative similarity of the mean community composition, the temporal variance of community composition within each set of parameters, $\psi$, provided a complementary perspective to investigate disturbance effects on benthic dynamics. We characterized the temporal variance of the coral assemblage by calculating, for each simulation, the variance over the entirety of simulation time of the fractional area cover. We define the variance, $v_c$, as the summed square of deviation from the long-term mean:

$$v_c = \frac{\sum_{t=1}^{t_{max}} [(C_{T,t} - \bar{C}_T)^2 + (C_{D,t} - \bar{C}_D)^2 + (C_{M,t} - \bar{C}_M)^2]}{t_{max} - 1}, \qquad (2.7)$$

with $\bar{C}_j$ being the mean fractional area covered through time by the coral belonging to type $j$, and $t_{max}$ the total simulation time (here 500 years). The relative simplicity of the five-parameter mortality representation highlights our interest in understanding the evolution of coral community structure over a wide range of disturbance scenarios (and associated mortality impacts) of differing intensity and probability. Our goal is not to simulate the dynamics of a set of specific reef locations, rather to explore in a theoretical framework the influence of mortality scenarios on coral community composition and variance. Nevertheless, our extensive exploration of realistic mortality regimes (100 000 scenarios) leads to the simulation of a vast array of environmental contexts under current, but also potentially future, disturbance regimes. All simulations and analysis were done using R v. 3.4.0.

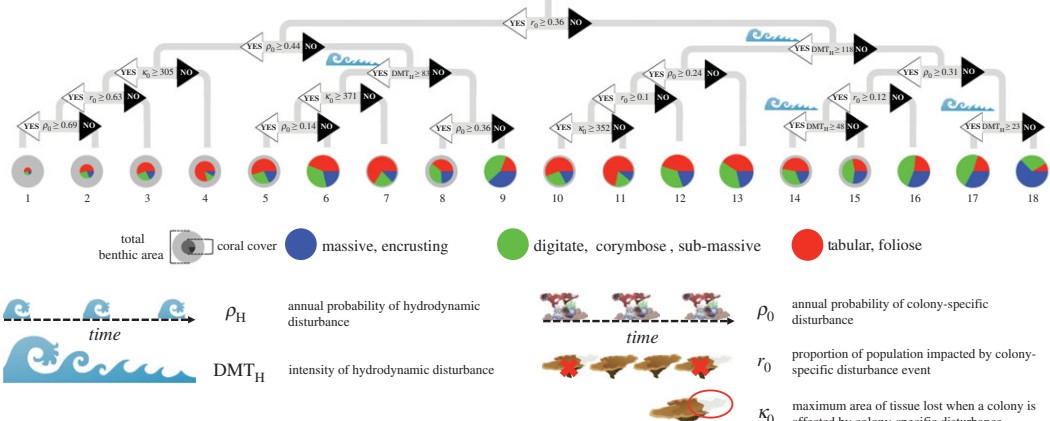

**Figure 1.** Summary of outputs from a multivariate regression tree analysis, summarizing the structuring of 100 000 simulations across ranges of five disturbance parameters. A total of 17 parameter conditions result in 18 stereotypical coral communities. Range of parameter values explored are presented in table 1. Each parameter condition presented here is hierarchical, structured as a decision tree with parameter inequality noted in the break, agreement with inequality labelled with a white arrow (and text 'YES', always linked to following tree to the left), and disagreement labelled with a black arrow (and text 'NO', always linked to following tree to the right). Only four parameter conditions in the final tree are dependent on hydrodynamic parameters, and each is labelled with an inset outline of disturbance icon. Stereotypical assemblages are the mean composition of the coral assemblage over 500 years of simulation and are represented by a pie chart with wedge colour corresponding to the proportion of each coral morphological group (tabular/foliose in red; digitate/corymbose/sub-massive in green and the encrusting/massive in blue); the size of the inner coloured circle represents the total coral area relative to total benthic area (light grey background circle) available to coral recruitment and growth.

## 3. Results

Regression tree analysis resulted in a binary tree containing 18 terminal classes of coral assemblage structure typical of each nested sequence of parameter values characterizing the type of disturbance experienced by the assemblage (figure 1). Notably, change in the probability of experiencing a hydrodynamic disturbance ($\rho_H$) did not affect coral assemblage composition; this parameter was not influential in determining any of the 18 terminal stereotypical assemblages (figure 1). By contrast, the intensity of hydrodynamic events ($DMT_H$) contributed to multiple splits in the classifications. The stereotypical assemblages (i.e. the mean of mean assemblages contained in each terminal class) exceeded 25% cover of massive/encrusting only in conditions of low $DMT_H$ (<117.7). Recall that low values of $DMT_H$ are linked with high susceptibility of individual corals to whole-colony mortality associated with wave activity (equation (2.4)) [27]. Digitate/corymbose corals also showed some evidence of release under high intensity of hydrodynamic disturbance, with some of the highest relative cover values of this type found when $DMT_H$ was low (figure 1, assemblages 9, 15, 16 and 17). These results indicate that across magnitudes of hydrodynamically forced mortality events, the regenerative capabilities of coral populations (across the spectrum of realized growth rates) can compensate largely for areal loss of corals on time scales shorter than the period between such events.

Changes in parameters associated with colony-specific disturbances were found to define 13 of the branches in our regression tree. When the proportion of the population impacted by colony-specific disturbance reaches or exceeds 36% ($r_0 \geq 0.36$) and the annual probability of such disturbances reaches or exceeds 44% ($\rho_0 \geq 0.44$), the intensity of hydrodynamic disturbances has no effect on terminal classification (figure 1, assemblages 1–4). These high-impact, high-frequency colony-specific disturbances lead to relatively low total coral cover (10–50%) dominated by tabular/foliose corals (greater than 50% of total coral cover). Tabular/foliose corals occupy more than 50% of the relative coral cover only under conditions when the proportion of population affected by a colony-specific disturbance exceeds 10% ($r_0 \geq 0.10$); recall that tabular/foliose have the highest growth rate of coral morphologies. The observation that assemblage structure under intense local-scale disturbance (i.e. disturbances associated with colony-specific disturbance) becomes unrelated to oceanographic conditions (e.g. disturbances associated with hydrodynamic conditions) and dominated by rapidly-growing taxa is consistent with empirical observations showing similar hydrodynamic disconnect in severely anthropogenic-impacted reefs [3].

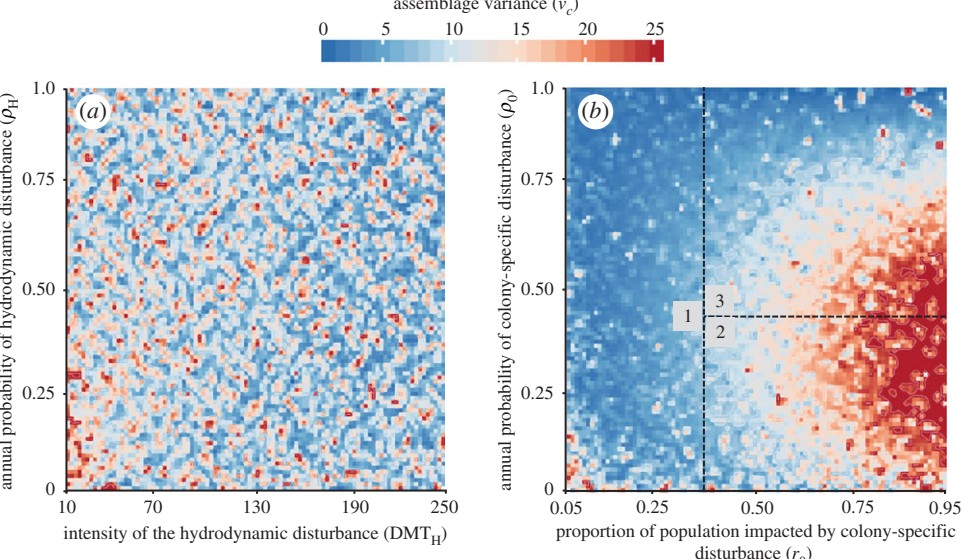

**Figure 2.** Heat map of variance in assemblage structure, summarizing the results of 100 000 simulations across ranges of two disturbance parameters. The same data are presented in panels (*a*) and (*b*), differing solely in the disturbance parameters across which the variance data are presented, across two parameters describing hydrodynamic disturbance and colony-specific disturbance, respectively. (*a*) Estimated temporal variance ($v_c$) of the coral communities as a function of the intensity of the hydrodynamic disturbance ($DMT_H$) and the annual probability of a hydrodynamic disturbance ($\rho_H$). (*b*) Estimated temporal variance ($v_c$) of the coral communities as a function of the proportion of population impacted by colony-specific disturbance ($r_0$) and the annual probability of colony-specific disturbance event ($\rho_0$). Patterning in the variance is related to two major splits in the regression tree (figure 1). The first split in the regression tree can be visualized as zone 1 (depicted with the number '1' in panel *b*), defined by the condition $r_0 < 0.36$, which contains simulations leading to stereotypical assemblages 10–18 in figure 1. Within the range $r_0 \geq 0.36$, the second split maps onto the variance; zone 2 ($\rho_0 < 0.44$) contains simulations leading to stereotypical assemblages 5–9 in figure 1 and the remainder in zone 3 contains simulations leading to stereotypical assemblages 1–4 in figure 1.

In addition to mean assemblage structure, the fluctuations of coral assemblage structure through time are influenced in a complementary way by rates and intensities of disturbances. When quantified as the mean squared Euclidean distances from the multivariate mean through time, assemblages spanned a 100-fold range of variance values (figure 2), from assemblages deviating little from the long-term mean to those exhibiting broad and frequent deviations. Across all simulations, the probability and intensity of hydrodynamic disturbance had an inconsistent paired influence on long-term variance (note lack of clustering in variance intensity across parameter space (figure 2*a*)). By contrast, changes in the probability of colony-specific disturbance and the proportion of the community affected by those disturbances, led to systematic effects in the variance of stereotypical assemblages through time (figure 2*b*).

The temporal variance of coral assemblages reached a maximum when many individuals within the coral population suffer effects of colony-specific disturbance at intermediate frequencies (figure 2*b*). The threshold values defining two of the first three most explanatory splits of the tree, the first common split ($r_0 = 0.36$) and one second split ($\rho_0 = 0.44$, defining assemblages 1–9, figure 1), map informatively upon the region of maximum temporal variance of assemblage structure (saddle region in $r_0 > 0.36$ and split by $\rho_0 = 0.44$; figure 2*b*). A 'ridge' of maximum variance exists when colony-specific disturbance exceeds 36%, with the variance showing a maximum when the annual probability of colony-specific disturbance events nears 44% and dropping at higher and lower values of $\rho_0$ (figure 2*b*). While these parameter thresholds for colony-specific disturbance events are critical in determining final assemblage structure (figure 1), they also reveal temporal patterns associated with thresholds among these groups of terminal classes (figure 2).

## 4. Discussion

Coral reefs, like most communities, exist in a perpetual state of compositional change owing to the impacts of episodic disturbance events [49–52]. The number, size and distribution of organisms are linked to the

history of disturbances and the biological mechanisms of death, recruitment and growth [45,53–55]. For many communities of sessile organisms, changes induced by disturbances can include contrasting patterns of mortality within and across species, ranging from complete loss of the individual through to partial mortality [56]. The modular nature of reef-building corals allows for a particularly wide range of responses to disturbance. Partial colony mortality provides a robust mechanism to suffer disturbance-induced insults but maintain colony-specific perpetuity through partial survival and (in some cases) subsequent regrowth [57,58]. We use a numerical modelling context to explore how life-historical trade-offs among reef-building corals, especially linked with particulars of colony morphology and related patterns of partial- and whole-colony mortality, can link environmental disturbance scenarios to the emergent structure of the coral assemblage. We explore the 'universe' of possible and likely environmental disturbance scenarios, parametrized based upon ranges of observed forcing conditions (table 1). The range of parameters explored describes the myriad combinations of conditions that characterize the extreme diversity of environmental conditions in which coral reefs exist across the globe.

Our findings suggest that the structure of a coral community is less influenced by impacts of episodic disturbances that are explicitly linked to loss of entire colonies, and is more influenced by scattered but ubiquitous disturbances that result in a variable amount of partial mortality in individual colonies (associated with myriad colony-specific disturbances, e.g. coral bleaching, disease, predation). We consider a spectrum of disturbances based on empirically-founded parameterizations associated with each hydrodynamic disturbances (that deterministically affect morphologically vulnerable colonies through whole-colony mortality owing to dislodgement) and colony-specific disturbances (that stochastically affect all colonies and are associated with variable amounts of areal loss, from partial to whole-colony mortality). Even though intense wave events can shift assemblage structure through removal of entire colonies, coral community structure reveals little association with the frequency of hydrodynamic disturbance events across realistic parameter ranges (figure 1). Clearly, there are short-term changes in coral communities following individual wave events, yet when averaged over decadal scales, the frequency of big wave events has only minor impact on coral assemblage structure. While size structure may shift, the relative abundance of coral morphological types remains consistent. Contributing to the community-structural resistance to frequency of big wave events is the patterning of growth rates across coral morphologies—the coral morphology most vulnerable to whole-colony mortality associated with big waves, the tabular, contain among the most rapidly growing taxa (table 1) [38,39].

Unlike frequency, the intensity of hydrodynamic disturbance events, which is linked to the proportion of the population affected by a given disturbance, has a quantifiable impact on the emergent structure of the coral assemblage. Stony corals have direct and predictable relationships between whole-colony dislodgement as a function of wave energy [27]. The functional relationship is influenced by the shape of the coral, both in terms of the colony shape (i.e. growth morphology) and size [28]. A notable pattern revealed in this study was the influence of intensity of hydrodynamic disturbance and the relative representation of massive corals; with higher intensity of wave events, massive corals became more represented (see branches with lower values of $DMT_H$ (higher intensity) in figure 1). Although massive corals tend to have the slowest growth rates among corals, the competitive opportunities for their success appear to include high intensity hydrodynamic conditions, in which the low-profile morphology increases resistance to hydrodynamic dislodgement [59].

The quantitative details of colony-specific disturbance events are more closely linked to the emergent structure of the reef community. Such mortality events, which can affect corals of all morphologies and all sizes similarly, tend to promote the dominance of fast-growing corals, in particular those with tabular/foliose morphology (assemblages 1–5, 7, 10, 11, figure 1). This reduction of life-historical diversity could decrease the ecological resilience of the coral reef ecosystem [60]. When the probability of colony-specific disturbances exceeds apparent thresholds, not only does the relative diversity of coral morphologies change, but total coral cover suffers appreciable reduction (assemblages 1–4, 14, 15, figure 1). In a comparison of reef benthic structure between unpopulated and populated islands of the Pacific, the reefs from populated islands had lower coral cover [2] and less correlation with hydrodynamical forcing [3]. Given that colony-specific disturbances may be higher on most inhabited reefs (e.g. disease, coral vulnerability to thermal stress), these empirical observations are consistent with the findings presented here. The loss of coral coverage can lead to major socio-ecological consequences, including reduced structural complexity, reductions in habitat availability for fishes and reductions in shoreline protection [61,62].

Complementing the analysis of the mean stereotypical assemblage structure, the relative magnitude of variance around these attractors provides novel insights into the dynamical distinctions between assemblages with qualitatively similar mean compositions. Inspection of figure 1 suggests that

multiple groups of stereotypical assemblages fluctuate around almost indistinguishable proportional composition of coral types (e.g. stereotypical assemblages 6 versus 12, 7 versus 11, 9 versus 17, figure 1). However, patterns of variance are generally different across these pairs, with stereotypical assemblages 5–9 lying within a zone of relatively high variance (zone 2, figure 2b) and stereotypical assemblages 10–18 lying within a zone of low variance (zone 1, figure 2b). Further, the stereotypical assemblages with the consistently lowest total coral cover (stereotypical assemblages 1–4, figure 1) are associated with the steepest gradient of variance (zone 3, figure 2b). These shifts are consistent with hypotheses of adaptive capabilities of benthic assemblages in response to changes in physical forcing. Specifically, with infrequent and low intensity effects of colony-specific disturbance, the assemblage shows little temporal variance consistent with rapid recovery towards the long-term mean assemblage structure (stereotypical assemblage 10–18, figure 1; zone 1, figure 2b). If the colony-specific disturbance results in impacts of high intensity and low frequency, then the assemblage will recover, but more slowly and with increased temporal variance (stereotypical assemblage 5–9, figure 1; zone 2, figure 2b). Comparable increases in state-space variance has been viewed as a harbinger of dramatic transitions in simple systems [63]; when the maximum amount of temporal variance is reached, the assemblage crosses a tipping point from which any other increase in intensity or frequency results in a decrease in temporal variance and the assemblage moves to more rapidly evolving, 'early successional' stage of assemblage development (e.g. stereotypical assemblage 1–4, figure 1; zone 3, figure 2b). Despite evidence for such tipping points in relation to colony-specific disturbance, neither the intensity nor the probability of hydrodynamic disturbance exhibits such systematic relation to the temporal variance of the assemblage (figure 2a).

Finding means to generalize ecological transitions on coral reefs is challenged by the notable high biodiversity in this system. While hydrodynamic disturbances can be mapped clearly onto a coral's morphology [27], establishing robust linkages between morphology and susceptibility to colony-specific disturbances proves to be less obvious. Colony-specific disturbances, as defined here, can be seen to represent, but are not limited to, thermal anomalies and exposure to coral diseases or corallivorous predators. Case studies exist of morphologically specific responses to some disturbances. For example, during a warm-water event in Okinawa, branching and digitate corals, the acroporids and pocilloporids, were disproportionately lost relative to massive and encrusting taxa [13]. However, syntheses of the standardized bleaching response revealed that bleaching impacts are often context-dependent and show little consistency across groups of taxa; genera dominated by each morphological group (i.e. massive, digitate or tabular) were not clearly distinguishable in ranges of observed bleaching responses (see fig. 3 in [32]). Analogous context-dependent patterning exists for other forms of colony-specific disturbances such as disease and predation by corallivores, which show some taxonomic bias in infection susceptibility and feeding preference, respectively [29]. However, there is limited evidence that such taxonomic bias maps consistently into morphological bias in mortality effects across taxa [43,64–66]. Taken across contexts, the effects of non-hydrodynamic disturbances among corals show quantitative links to taxonomy of the coral, but these effects show at most weak linkages to the morphology of the corals. However, in cases where patterns of colony-specific mortality can be mapped clearly onto morphological types of corals, a significant research opportunity exists to extend specific applications of more precisely parametrized disturbance models, such as presented here, to build predictions of community composition and patterns of variability through time.

The variance of coral reef benthic community structure, both through time and across geography, challenges ecological approaches to build quantitative models of composition. Collections of efforts have sought to describe behaviour of simplified systems, for example competition among stereotyped scleractinian corals and coarse algal types [33,34,67]. Finding means to explore shifts in coral community structure with higher resolution depends upon a balanced approach of increasing model complexity while maintaining analytical parsimony. Use of morphological groups in an exploration of coral community responses to physical and biological disturbances provides one avenue for generating predictions. Importantly, the increase in model complexity introduces additional challenges for exploring model output. The use of computationally intensive, yet theoretically transparent, methods of statistical clustering, i.e. regression tree analysis, is a powerful approach for producing generalizations of output and behaviour among increasingly parameter-heavy models [68].

In this study, we find that the structure of a coral community can be influenced strongly by patterns of disturbance. Hydrodynamic disturbances, as driven often by the frequency and intensity of big wave events, have targeted effects on coral community structure. Interestingly, such disturbances are in many cases limited in their medium-term effects owing to the relatively rapid regrowth capacity of coral taxa. By contrast, colony-specific disturbances can have much more ubiquitous and profound

impacts on the structure of reef communities. Such disturbances can have a range of impacts on individual corals, ranging from partial colony mortality for larger colonies to whole-colony loss for smaller colonies. Further, such disturbances tend to occur more frequently than big-wave events (table 1), thus overwhelming the quantitative rates of regrowth for all but a few coral taxa. Even when corals can maintain community structure, the temporal variance around the long-term mean can increase (figure 2), making empirical tracking of community structure more challenging owing to short time-scale fluctuations. Coral communities are variable, yet we can develop robust and testable predictions linking disturbance patterns to expected community structure. As growing collections of long-term data emerge, such predictions serve as guides for exploring patterns of variation and change across natural coral reef seascapes.

Data accessibility. Data and relevant code for this research work are stored in GitHub: https://github.com/clinton-edwards/Disturbance_impacts_on_coral_assemblages and have been archived within the Zenodo repository: https://zenodo.org/record/3975938.

Authors' contributions. S.A.S. and Y.E. designed and implemented the model and ran the simulations. S.A.S., Y.E., C.B.E., G.J.W. and D.E.M. analysed the model output and wrote the manuscript. All authors gave final approval for publication.

Competing interests. We declare we have no competing interests.

Funding. This work is a contribution of the Reefs Tomorrow Initiative, a programme funded by the Gordon and Betty Moore Foundation (grant no. 3420).

Acknowledgements. The authors want to thank Jennifer E. Smith and David Nerini for their valuable comments and suggestions.

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
