## [Reviewer comments · Royal Society Open Science]

Review History

RSOS-200565.R0 (Original submission)

Review form: Reviewer 1

Is the manuscript scientifically sound in its present form?

Yes

Are the interpretations and conclusions justified by the results?

Yes

Is the language acceptable?

Yes

Do you have any ethical concerns with this paper?

No

Have you any concerns about statistical analyses in this paper?

No

Recommendation?

Accept with minor revision (please list in comments)

Comments to the Author(s)

See attachment (Appendix A).

Review form: Reviewer 2**Is the manuscript scientifically sound in its present form?**

Yes

Are the interpretations and conclusions justified by the results?

Yes

Is the language acceptable?

Yes

Do you have any ethical concerns with this paper?

No

Have you any concerns about statistical analyses in this paper?

No

Recommendation?

Accept with minor revision (please list in comments)

Comments to the Author(s)

This is really quite a nice paper. Modeling is clean and makes some nice points. Writing is clear. Figure 1 is a really great breakdown of the relative roles of the different forms of disturbance. Figure 2 very nice and well mapped to figure 1.

I have two high-level comments:

First: The authors have a generic “colony specific” form of disturbance, which they argue represents bleaching, thermal stress, coral disease, predators, maybe others. But I suspect this was not actually intended to represent those processes, but was rather conceived of as a generic form of disturbance for comparison with the hydrodynamic disturbances which were (I suspect) the original focus of the modeling.

Reading between the lines, it seems that the authors set out to explore the role of hydrodynamic disturbance. They do a nice job of structuring the model to capture some key elements of this disturbance in terms of how it affects different colony sizes differently and how this scaling differs as a function of morphology. And then for comparison they have also developed a second form of disturbance which does not depend on morphology or taxa, and has impacts which are often sub-lethal. This second form of disturbance seems to have been intended as a catchall comparison rather than a detailed representation of a particular disturbance.

The complication for the authors is that they have found that the second, more generic disturbance is actually much more important to community structure than the hydrodynamic disturbance which was the actual focus of their research plan. This creates a challenge for storytelling because they have to focus on the effects of the generic disturbance but it is not really well mapped on a particular real process.

Unfortunately, this storytelling challenge has tempted the authors into pushing the idea that this generic form of disturbance is really the correct way to represent effects from...well they kind of imply all non-hydrodynamic disturbances.... But they list bleaching, thermal stress, coral disease,

predators. There are some arguments in the discussion for example around line 408 that such non-physical disturbances have not been shown to have consistently disparate effects on different taxa. But this seems a pretty weak. Bleaching events often affect only a subset of taxa on a reef, with effects that are consequently concentrated on certain morphotypes. Perhaps there are no globally reliable relationships, but I am a little suspicious that the lack of such relationships is a consequence of their lumping so many disturbances together. If they initiated a modeling study looking only a crown of thorns starfish, I expect they would have constructed a model of that disturbance which differs in how it varies with size and morphotype than if they constructed a model of bleaching disturbance. I am not arguing that they redo the modeling and add more disturbances, only that they acknowledge that there is probably more to learn from more focused models of the disturbances that they lump together here (and that their results point towards that as important next steps for such work). I note a few specific places where they ought to dial back their claims below.

Second: I found the abstract and intro hard to follow. Did not really understand what was being modeled until the methods. I expect that is partly because of the storytelling complexity discussed above. Lines 22-23 should more clearly state what is being modeled (not environmental forcing which could mean anything). And just in general they should accept their main results and state them more directly:

- hydrodynamic disturbance surprisingly unimportant for community structure
- other disturbance regimes potentially very important; more work needed on those sources

Minor points:

Line 27: Acknowledge that this is not really a model of coralivory, disease or stress. 24-26 do a nice job of stating negative hydrodynamic results, line 27 should just say that more generic disturbances can be strong drivers of morphotype composition. (I am not clear that you showed that “diffusive” or “localized” are the key elements of this generic disturbance, as many factors are different from the hydrodynamic disturbance).

Line 43: where? This sentence seems oddly elliptical.

Line 88-101: This model is really cool and nicely described.

Line 102-116: also well described, but I have a little trouble what makes these “colony specific”. The hydrodynamic disturbance is also “colony specific” in that you flip a coin for each colony with weighting based on disturbance intensity and colony size/morphotype? I get the sense that the two disturbance types are different in terms of how correlated the disturbances are across the landscape... can you show that somehow? Perhaps in another figure?

Line 124: Growth is proportional to free space in the model. Is this a good approximation given the geometry of real corals? Does (total space – all coral) really map linearly on (opportunities for growth)? I know the Sandin lab has pushed as far as anyone on these kinds of questions so defer to their judgement, but perhaps a sentence addressing this would be appropriate

Line 159-160: As discussed above, I am not convinced of this statement for the real world. Perhaps say “ In this model all episodic disturbances...”

Line 204-205: Fine to say this is beyond the scope, but I would not expect to have alternate states in this model (I do not see any relevant positive feedbacks). You introduce the idea that there might be alternative attractors. Do you see evidence of this? If not might just say so here.

Line 208: redundant but not consistent with 200.

Line 224-236: super cool approach. But the tree is just based on means not variance right? That is clear here but gets confusing later.

Figure 1 is magnificent. But the embedded icons are a little confusing. Could they be placed to show which side has more wave disturbance? As it is they are inconsistent as to which side they are on.

Line 329-330: Is the partial-mortality the key part here?

Line 349-352: this is quite a nice explanation.

Line 355-356: Just seems too strong a statement. A bunch of assumptions were made here about how the colony specific disturbances scale with colony size (and do not scale with morphology). Assumptions are perfectly reasonable choices but more focus on an individual disturbance (i.e. bleaching) might find literature supporting different choices.

Line 369-376: super interesting stuff but hard to get this from the figures. Current mapping is very good but does not let one compare assemblage 6 vs 12 (for example). Could each assemblage be mapped on its own (partial) version of figure 2 . Like an 18 panel supplement? Or maybe something cleverer? In any case I think maybe this part could be introduced in the results?

Line 394-412: The fact that local patterns of morphologically specific loss do not hold up in global analyses does not mean these disturbances are not morphologically biased at local scales. I do not find this a convincing argument that the particular choices made in the "colony-specific" disturbances modeled here are more valid than other choices.

Review form: Reviewer 3 (Peter Mumby)

Is the manuscript scientifically sound in its present form?

Yes

Are the interpretations and conclusions justified by the results?

Yes

Is the language acceptable?

Yes

Do you have any ethical concerns with this paper?

No

Have you any concerns about statistical analyses in this paper?

No

Recommendation?

Accept with minor revision (please list in comments)

Comments to the Author(s)

This is a useful paper that contrasts the impacts of system-wide physical disturbance upon corals from 'biological' processes that vary considerably among taxa and with size. A novel aspect of the paper is the categorisation of emergent community state using a clustering approach to link disturbance regime to eventual outcome.

The rationale of the study is described clearly. What I found less clear was the parameterisation of non-physical biological disturbances. I agree that it makes sense to generate a broad parameter space of partial and whole-colony mortality and examine how these influence ecological

outcomes but I was left wondering whether some of these scenarios are more likely than others. The description of model parameters does not reveal which (if any) of the biological stresses are based on empirical data. Equally, it would be helpful to consider whether some of the outcomes are more representative of specific disturbance scenarios.

Some differences among taxa can be obtained from Ortiz et al 2014 (Nature Climate Change).

But overall this is a useful contribution to the disturbance literature.

Decision letter (RSOS-200565.R0)

Dear Dr Sandin,

The editors assigned to your paper ("Modelling the linkage between coral assemblage structure and pattern of environmental forcing") have now received comments from reviewers. We would like you to revise your paper in accordance with the referee and Associate Editor suggestions which can be found below (not including confidential reports to the Editor). Please note this decision does not guarantee eventual acceptance.

Please submit a copy of your revised paper before 05-Aug-2020. Please note that the revision deadline will expire at 00.00am on this date. If we do not hear from you within this time then it will be assumed that the paper has been withdrawn. In exceptional circumstances, extensions may be possible if agreed with the Editorial Office in advance. We do not allow multiple rounds of revision so we urge you to make every effort to fully address all of the comments at this stage. If deemed necessary by the Editors, your manuscript will be sent back to one or more of the original reviewers for assessment. If the original reviewers are not available, we may invite new reviewers.

- Data accessibility

<http://datadryad.org/submit?journalID=RSOS&manu=RSOS-200565>

- Competing interests

- Authors' contributions

- Acknowledgements

- Funding statement

Kind regards,

Andrew Dunn

Comments to Author:

Reviewers' Comments to Author:

Reviewer: 1

Comments to the Author(s)

See attachment

Reviewer: 2

Comments to the Author(s)

This is really quite a nice paper. Modeling is clean and makes some nice points. Writing is clear. Figure 1 is a really great breakdown of the relative roles of the different forms of disturbance. Figure 2 very nice and well mapped to figure 1.

I have two high-level comments:

First: The authors have a generic "colony specific" form of disturbance, which they argue represents bleaching, thermal stress, coral disease, predators, maybe others. But I suspect this was not actually intended to represent those processes, but was rather conceived of as a generic form of disturbance for comparison with the hydrodynamic disturbances which were (I suspect) the original focus of the modeling.

Reading between the lines, it seems that the authors set out to explore the role of hydrodynamic disturbance. They do a nice job of structuring the model to capture some key elements of this disturbance in terms of how it affects different colony sizes differently and how this scaling differs as a function of morphology. And then for comparison they have also developed a second form of disturbance which does not depend on morphology or taxa, and has impacts which are often sub-lethal. This second form of disturbance seems to have been intended as a catchall comparison rather than a detailed representation of a particular disturbance.

The complication for the authors is that they have found that the second, more generic disturbance is actually much more important to community structure than the hydrodynamic disturbance which was the actual focus of their research plan. This creates a challenge for storytelling because they have to focus on the effects of the generic disturbance but it is not really well mapped on a particular real process.

Unfortunately, this storytelling challenge has tempted the authors into pushing the idea that this generic form of disturbance is really the correct way to represent effects from...well they kind of imply all non-hydrodynamic disturbances.... But they list bleaching, thermal stress, coral disease, predators. There are some arguments in the discussion for example around line 408 that such non-physical disturbances have not been shown to have consistently disparate effects on different taxa. But this seems a pretty weak. Bleaching events often affect only a subset of taxa on a reef, with effects that are consequently concentrated on certain morphotypes. Perhaps there are no globally reliable relationships, but I am a little suspicious that the lack of such relationships is a consequence of their lumping so many disturbances together. If they initiated a modeling study looking only a crown of thorns starfish, I expect they would have constructed a model of that disturbance which differs in how it varies with size and morphotype than if they constructed a model of bleaching disturbance. I am not arguing that they redo the modeling and add more disturbances, only that they acknowledge that there is probably more to learn from more focused models of the disturbances that they lump together here (and that their results point towards that as important next steps for such work). I note a few specific places where they ought to dial back their claims below.

Second: I found the abstract and intro hard to follow. Did not really understand what was being modeled until the methods. I expect that is partly because of the storytelling complexity discussed above. Lines 22-23 should more clearly state what is being modeled (not

environmental forcing which could mean anything). And just in general they should accept their main results and state them more directly:

- hydrodynamic disturbance surprisingly unimportant for community structure
- other disturbance regimes potentially very important; more work needed on those sources

Minor points:

Line 27: Acknowledge that this is not really a model of coralivory, disease or stress. 24-26 do a nice job of stating negative hydrodynamic results, line 27 should just say that more generic disturbances can be strong drivers of morphotype composition. (I am not clear that you showed that “diffusive” or “localized” are the key elements of this generic disturbance, as many factors are different from the hydrodynamic disturbance).

Line 43: where? This sentence seems oddly elliptical.

Line 88-101: This model is really cool and nicely described.

Line 102-116: also well described, but I have a little trouble what makes these “colony specific”. The hydrodynamic disturbance is also “colony specific” in that you flip a coin for each colony with weighting based on disturbance intensity and colony size/morphotype? I get the sense that the two disturbance types are different in terms of how correlated the disturbances are across the landscape... can you show that somehow? Perhaps in another figure?

Line 124: Growth is proportional to free space in the model. Is this a good approximation given the geometry of real corals? Does (total space – all coral) really map linearly on (opportunities for growth)? I know the Sandin lab has pushed as far as anyone on these kinds of questions so defer to their judgement, but perhaps a sentence addressing this would be appropriate

Line 159-160: As discussed above, I am not convinced of this statement for the real world. Perhaps say “ In this model all episodic disturbances...”

Line 204-205: Fine to say this is beyond the scope, but I would not expect to have alternate states in this model (I do not see any relevant positive feedbacks). You introduce the idea that there might be alternative attractors. Do you see evidence of this? If not might just say so here.

Line 208: redundant but not consistent with 200.

Line 224-236: super cool approach. But the tree is just based on means not variance right? That is clear here but gets confusing later.

Figure 1 is magnificent. But the embedded icons are a little confusing. Could they be placed to show which side has more wave disturbance? As it is they are inconsistent as to which side they are on.

Line 329-330: Is the partial-mortality the key part here?

Line 349-352: this is quite a nice explanation.

Line 355-356: Just seems too strong a statement. A bunch of assumptions were made here about how the colony specific disturbances scale with colony size (and do not scale with morphology). Assumptions are perfectly reasonable choices but more focus on an individual disturbance (i.e. bleaching) might find literature supporting different choices.

Line 369-376: super interesting stuff but hard to get this from the figures. Current mapping is very good but does not let one compare assemblage 6 vs 12 (for example). Could each

assemblage be mapped on its own (partial) version of figure 2 . Like an 18 panel supplement? Or maybe something cleverer? In any case I think maybe this part could be introduced in the results?

Line 394-412: The fact that local patterns of morphologically specific loss do not hold up in global analyses does not mean these disturbances are not morphologically biased at local scales. I do not find this a convincing argument that the particular choices made in the “colony-specific” disturbances modeled here are more valid than other choices.

Reviewer: 3

Comments to the Author(s)

This is a useful paper that contrasts the impacts of system-wide physical disturbance upon corals from 'biological' processes that vary considerably among taxa and with size. A novel aspect of the paper is the categorisation of emergent community state using a clustering approach to link disturbance regime to eventual outcome.

The rationale of the study is described clearly. What I found less clear was the parameterisation of non-physical biological disturbances. I agree that it makes sense to generate a broad parameter space of partial and whole-colony mortality and examine how these influence ecological outcomes but I was left wondering whether some of these scenarios are more likely than others. The description of model parameters does not reveal which (if any) of the biological stresses are based on empirical data. Equally, it would be helpful to consider whether some of the outcomes are more representative of specific disturbance scenarios.

Some differences among taxa can be obtained from Ortiz et al 2014 (Nature Climate Change).

But overall this is a useful contribution to the disturbance literature.

Author's Response to Decision Letter for (RSOS-200565.R0)

See Appendix B.

RSOS-200565.R1 (Revision)

Review form: Reviewer 1

Is the manuscript scientifically sound in its present form?

Yes

Are the interpretations and conclusions justified by the results?

Yes

Is the language acceptable?

Yes

Do you have any ethical concerns with this paper?

No

Have you any concerns about statistical analyses in this paper?

No

Recommendation?

Accept as is

Comments to the Author(s)

I'm fully satisfied with the authors' response and edits to the MS and congratulate them with this excellent piece of work!

Review form: Reviewer 2

Is the manuscript scientifically sound in its present form?

Yes

Are the interpretations and conclusions justified by the results?

Yes

Is the language acceptable?

Yes

Do you have any ethical concerns with this paper?

No

Have you any concerns about statistical analyses in this paper?

No

Recommendation?

Accept as is

Comments to the Author(s)

This revision is really well done. Paper is great and ready for publication.

The adjustments to the abstract and introduction have really clarified the main points about disturbance. I they retain their arguments on pages 4 and 6 about why the "generic disturbance" is a good representation of disturbances cause by bleaching disease etc, but these make more sense and feel appropriate in the context of the revised framing. It also helps that there is more clarity of the importance of partial vs whole colony mortality, which was in the previous draft but not as clearly stated.

I accept the author's arguments why the discussion of variance starting around lines 395 needs to go into the discussion. And I take their point that another figure would be hard to add without a lot of redundancy. I just wish it was easier for the reader to connect the stereotypical assemblages from figure 1 to the associated variance results on figure 2. I note the reader can use the regression tree to figure out where each assemblage is and then scribble them in, and my attempt to do that was....well I see how it is hard to visualize :)

Review form: Reviewer 3 (Peter Mumby)

Is the manuscript scientifically sound in its present form?

Yes

Are the interpretations and conclusions justified by the results?

Yes

Is the language acceptable?

Yes

Do you have any ethical concerns with this paper?

No

Have you any concerns about statistical analyses in this paper?

No

Recommendation?

Accept with minor revision (please list in comments)

Comments to the Author(s)

I appreciate the efforts the authors have gone to in revising their paper. I think it's almost publishable but my comment 'are some areas of parameter space more likely than others' doesn't seem to have been addressed. I have read the responses to other referees but am still left wondering which areas of parameter space are more realistic or indicative of particular scenarios in situ. The response argues that the language has been softened and that readers are directed to consider more nuanced studies of colony morphology and that these should be modelled specifically in future. Yet the abstract makes strong conclusions about the lack of influence of hydrodynamic conditions. To me this doesn't reconcile. If you're going to make strong conclusions then I think you need to reflect on the degree to which your model will reflect real-life dynamics.

I still believe that the model is useful and appreciate the results - I'd just like to see more thought placed into what is really being represented here.

Decision letter (RSOS-200565.R1)

Dear Dr Sandin

On behalf of the Editors, we are pleased to inform you that your Manuscript RSOS-200565.R1 "Modelling the linkage between coral assemblage structure and pattern of environmental forcing" has been accepted for publication in Royal Society Open Science subject to minor revision in accordance with the referees' reports. Please find the referees' comments along with any feedback from the Editors below my signature.

Please submit your revised manuscript and required files (see below) no later than 7 days from today's (ie 16-Sep-2020) date. Note: the ScholarOne system will 'lock' if submission of the revision

is attempted 7 or more days after the deadline. If you do not think you will be able to meet this deadline please contact the editorial office immediately.

Associate Editor Comments to Author:

A few comments remain to be addressed, and we urge you to take these seriously in your final revision.

Reviewer comments to Author:

Reviewer: 1

Comments to the Author(s)

I'm fully satisfied with the authors' response and edits to the MS and congratulate them with this excellent piece of work!

Reviewer: 2

Comments to the Author(s)

This revision is really well done. Paper is great and ready for publication.

The adjustments to the abstract and introduction have really clarified the main points about disturbance. I they retain their arguments on pages 4 and 6 about why the "generic disturbance" is a good representation of disturbances cause by bleaching disease etc, but these make more sense and feel appropriate in the context of the revised framing. It also helps that there is more clarity of the importance of partial vs whole colony mortality, which was in the previous draft but not as cleanly stated.

I accept the author's arguments why the discussion of variance starting around lines 395 needs to go into the discussion. And I take their point that another figure would be hard to add without a lot of redundancy. I just wish it was easier for the reader to connect the stereotypical assemblages from figure 1 to the associated variance results on figure 2. I note the reader can use the regression tree to figure out where each assemblage is and then scribble them in, and my attempt to do that was....well I see how it is hard to visualize :)

Reviewer: 3

Comments to the Author(s)

I appreciate the efforts the authors have gone to in revising their paper. I think it's almost publishable but my comment 'are some areas of parameter space more likely than others' doesn't seem to have been addressed. I have read the responses to other referees but am still left

wondering which areas of parameter space are more realistic or indicative of particular scenarios in situ. The response argues that the language has been softened and that readers are directed to consider more nuanced studies of colony morphology and that these should be modelled specifically in future. Yet the abstract makes strong conclusions about the lack of influence of hydrodynamic conditions. To me this doesn't reconcile. If you're going to make strong conclusions then I think you need to reflect on the degree to which your model will reflect real-life dynamics.

I still believe that the model is useful and appreciate the results - I'd just like to see more thought placed into what is really being represented here.

===PREPARING YOUR MANUSCRIPT===

===PREPARING YOUR REVISION IN SCHOLARONE===

<https://royalsociety.org/journals/authors/author-guidelines/#supplementary-material> to include a suitable title and informative caption. An example of appropriate titling and captioning may be found at https://figshare.com/articles/Table_S2_from_Is_there_a_trade-off_between_peak_performance_and_performance_breadth_across_temperatures_for_aerobic_sc_ope_in_teleost_fishes_/3843624.

Author's Response to Decision Letter for (RSOS-200565.R1)

See Appendix C.

Decision letter (RSOS-200565.R2)

Dear Dr Sandin,

It is a pleasure to accept your manuscript entitled "Modelling the linkage between coral assemblage structure and pattern of environmental forcing" in its current form for publication in Royal Society Open Science.

Appendix A

This manuscript investigates the impact of global and local disturbances on absolute and relative coral cover of three coral morphotypes (sheets, staghorns and massives) with a simple modeling framework. It is well written, concise and to the point, the methodology is sound and the conclusions are well supported. There are a few things, though, that I think need to be taken care of before I would recommend the MS for publication.

1. The model implicitly assumes that coral reefs are invincible. Absolute coral cover may become very small and the relative abundance of morphotypes may be severely impacted by the types of disturbances considered in the study, but since there is always (density independent) recruitment and a total lack of competitors (CCA, macroalgae, etc.), survival of the three morphotypes is guaranteed. This gives an overly optimistic picture of the stability and resilience of reef systems. I'm anticipating the authors will reply that consideration of those is well beyond the scope of the paper, and I would wholeheartedly agree. But I'd suggest the authors include clear disclaimers in Abstract and Introduction and pick up the matter in the Discussion.

2. It appears that the model is pseudo-spatially explicit and follows the net growth of individual colonies in a 2D horizontal plane, but I'm not quite sure. Please clarify.

3. The authors state that the growth function (Equation 2) follows from the assumption that colonies grow circularly. What does that mean? Equation 2 has two terms denoting growth potential and a logistic space competition factor. The derivation of the first term is not immediately apparent and rests on the assumption that the absolute radial growth rate is constant (in absence of competition) for all morphotypes. I'd like to see some support for that assumption. In addition, the dimensions don't match - the g 's need to be multiplied by Δt or they need to be renamed to annual absolute radial growth (and the time dimension need to be removed from the Table). I think there is a typo in the logistic term. Shouldn't it be $C_{\{i,j,t-1\}}$ instead of $C_{\{i,j,t\}}$?

4. Recruits enter the population when they're about 11 cm in diameter. That's pretty large. In effect, actual recruits are sheltered from disturbance and do not affect the growth of larger colonies until they reach this size. The authors (l. 182) state that reducing the recruitment size doesn't change the results. This is counterintuitive to me, since colony specific loss rates decline with size (Eq. 5). Please explain (in the Discussion).

Minor points

Order entries of Table alphabetically.

I. 210-213 Don't really understand what is said here.

I. 206-222 Much of this seems to be Discussion material.

I. 229 pi has 2 meanings in the MS (and another one in the SI). Choose another symbol.

Figure 2A and 2B are mixed up at many (but not all) places in the Results and Discussion.

Appendix B

UNIVERSITY OF CALIFORNIA, SAN DIEGO

UCSD

BERKELEY • DAVIS • IRVINE • LOS ANGELES • MERCED • RIVERSIDE • SAN DIEGO • SAN FRANCISCO

SANTA BARBARA • SANTA CRUZ

SCRIPPS INSTITUTION OF OCEANOGRAPHY, 0202
CENTER FOR MARINE BIODIVERSITY AND CONSERVATION
TEL: (858) 822- 2790
FAX: (858) 822- 1267

9500 GILMAN DRIVE
LA JOLLA, CALIFORNIA 92093- 0202
EMAIL: CMBC@UCSD.EDU
URL: CMBC.UCSD.EDU

2 August 2020

Dear *RSOS* Editorial Team,

Thank you for securing thoughtful reviews, and for providing additional perspectives regarding our submission, "*Modelling the linkage between coral assemblage structure and pattern of environmental forcing*" (Ms. No. RSOS-200565). Following the editorial opportunity afforded, we are pleased to submit a revised manuscript for consideration by *Royal Society Open Science*.

We found the reviewers' comments to be encouraging and constructive. We appreciated the positive feedback regarding the framing of the questions, the statistical implementation, and the quality of writing and figures. We also appreciated the feedback regarding specific elements of the contextualization and definitions. The reviewers' commentary helped us to refine the manuscript; you will find three major updates in the revised manuscript.

- i. *Clarifying definitions and model assumptions.* The reviewers asked pointed questions regarding how we defined and justified a handful terms assumptions in the model. We have added justifications, or bolstered definitions for clarity, assuring that the model structure is transparent.
- ii. *Adding emphasis on finding that hydrodynamic disturbance has little impact.* Reviewer #2, in particular, highlighted that a strength of the manuscript is that hydrodynamic disturbance has relatively little effect on the assemblage structure of corals. Given that our defined 'colony-specific disturbance' has a more general interpretation than the precise definition of hydrodynamic disturbance, the reviewer suggests shifting some emphasis toward the latter. We agree with the suggestion, and the edits in that vein have strengthened the manuscript, in our opinion.
- iii. *Editorial specifics.* We appreciate the detailed read from the reviewers, as they identified a number of editorial omissions and errors. We have edited the manuscript, clearing up redundancies and the few typographical errors identified.

In the following pages, we provide more direct responses to editor and reviewer comments.

Thank you again for offering us an opportunity to submit this revised manuscript. We are very encouraged by the reviewers' enthusiasm and hope to share this work more broadly with the academic community soon.

Sincerely,

Stuart Sandin

Professor, Scripps Institution of Oceanography, UC San Diego

Cc: Yoan Eynaud, Gareth Williams, Clinton Edwards, and Dylan McNamara

Comments to authors from reviewers (with responses, **in bold**)

Reviewer: 1

This manuscript investigates the impact of global and local disturbances on absolute and relative coral cover of three coral morphotypes (sheets, staghorns and massives) with a simple modeling framework. It is well written, concise and to the point, the methodology is sound and the conclusions are well supported. There are a few things, though, that I think need to be taken care of before I would recommend the MS for publication.

We thank the reviewer for the overview, and for the constructive comments that follow.

1. The model implicitly assumes that coral reefs are invincible. Absolute coral cover may become very small and the relative abundance of morphotypes may be severely impacted by the types of disturbances considered in the study, but since there is always (density independent) recruitment and a total lack of competitors (CCA, macroalgae, etc.), survival of the three morphotypes is guaranteed. This gives an overly optimistic picture of the stability and resilience of reef systems. I'm anticipating the authors will reply that consideration of those is well beyond the scope of the paper, and I would wholeheartedly agree. But I'd suggest the authors include clear disclaimers in Abstract and Introduction and pick up the matter in the Discussion.

The reviewer has identified an important bound on the scope of this study. Indeed, we have not included reference to competition with non-coral benthic types, in essence simulating a coral reef benthic community where stony corals are the dominant space competitors (e.g., in the presence of an active community of herbivores where competition with fast-growing algal competitors is limited). And, yes, we felt that extending this analysis of relative performance of coral types could not be coupled with an analysis of competition with other benthic types without making the study too burdensome (thank you for the agreement on this point).

We have included some reference to the limits of this study in the Discussion originally, and following the recommendation of the reviewer we have added commentary to this point in the Abstract and Introduction. We agree that transparency on this point is critical for this study to contribute to the literature without creating confusion regarding the goals of this study. Our goal was to study how life history tradeoffs among corals can influence the composition of a coral assemblage across gradients of environmental forcings. We see shifting patterns of algal competition as a complement to these underlying coral-coral competitive shifts. [*Abstract has been largely re-crafted, and the last paragraph of the Introduction has been edited.*]

2. It appears that the model is pseudo-spatially explicit and follows the net growth of individual colonies in a 2D horizontal plane, but I'm not quite sure. Please clarify.

Yes, that is correct. The model simulates the growth of many individual corals, with the assumption that each individual is a circle in 2D space. We have added a clarifying sentence to introduce the model description. [Lines 132-133]

3. The authors state that the growth function (Equation 2) follows from the assumption that colonies grow circularly. What does that mean?

The design of the G term is based on the assumption that coral colonies are shaped as a circle in 2D space and grow radially, as stated in the text “We assume that the colonies grow circularly, and thus the growth function G is defined as...”. We did not however step through the derivation of that term to save space. For the reviewer, the G term comes about from the following assumptions,

$$C(t+1) = C(t) + G$$

$$G = C(t+1) - C(t)$$

$$C(t) = \pi * r(t)^2$$

$$C(t+1) = \pi * (r(t) + g)^2$$

G follows from solving the above.

Equation 2 has two terms denoting growth potential and a logistic space competition factor. The derivation of the first term is not immediately apparent and rests on the assumption that the absolute radial growth rate is constant (in absence of competition) for all morphotypes. I'd like to see some support for that assumption.

Multiple studies have supported the assumption of constant radial growth among coral taxa, including Hughes and Jackson (1985; *Ecological Monographs*) and more recently Dornelas et al (2017; *Proceedings of the Royal Society B*), the former showing data directly in units of radial growth (which is constant across sizes) and the latter showing allometric growth across multiple taxa of corals, which is consistent with constant radial growth. Beyond these studies, an exhaustive cataloging of coral growth rates does not exist to our knowledge. Given available evidence, we opted to start with the assumption that coral growth is constant given no competition and unlimited space. This assumption allows us to focus on the role that disturbance plays. We considered a temporally variable growth rate as an interesting avenue for future modeling work but for our contribution we do not feel as though our major results would be altered by a more complex growth rate. To the point of size-independent constant radial growth, we have added direct reference to the studies mentioned above in the text. [Lines 141-143]

In addition, the dimensions don't match - the g's need to be multiplied by delta t or they need to be renamed to annual absolute radial growth (and the time dimension need to be removed from the Table).

Sorry for the confusion. We have removed the time from the table.

I think there is a typo in the logistic term. Shouldn't it be $C_{\{i,j,t-1\}}$ instead of $C_{\{i,j,t\}}$?

Yes, thank you for alerting us to that typo.

4. Recruits enter the population when they're about 11 cm in diameter. That's pretty large. In effect, actual recruits are sheltered from disturbance and do not affect the growth of larger colonies until they reach this size. The authors (l. 182) state that reducing the recruitment size doesn't change the results. This is counterintuitive to me, since colony specific loss rates decline with size (Eq. 5). Please explain (in the Discussion).

The colony-specific loss rates do decline with size, and the growth rates (in units of area) increase with size. Across a range of smaller sizes, these contrasting rates work in opposite directions, thus leading to nominal quantitative impacts of altering definitions of recruitment. The recruitment rate, therefore, is a concatenation of early survival, early growth, and introduction of new individuals into the population. Of course, there is a minimum size where such flexibility of definitions breaks down, for example at the smallest sizes of coral settlers and similar. And, perhaps the principal reason that shifting this parameter value does not influence results is that the model works in a currency of coral area and adult corals have a disproportionately large contribution to shifts in area (through growth) than the small-radius recruits. We have introduced some additional text to clarify in the manuscript. We opted to include this text in the model introduction rather than the Discussion. [Lines 204-207]

Minor points

Order entries of Table alphabetically.

Changed.

1. 210-213 Don't really understand what is said here.

We have edited for clarity. [Lines 231-237]

1. 206-222 Much of this seems to be Discussion material.

We agree that much of this is superfluous to the Methods and has been deleted. [Lines 231-237]

1. 229 pi has 2 meanings in the MS (and another one in the SI). Choose another symbol.

We have changed the definition of variable space to a new symbol, here and in the SI (which is referencing this same variable space symbol). [Line 244 and SI]

Figure 2A and 2B are mixed up at many (but not all) places in the Results and Discussion.

Indeed, thank you for catching this. We have updated the labeling, including shifting the labeling in the Figure 2 itself.

Reviewer: 2

Comments to the Author(s)

This is really quite a nice paper. Modeling is clean and makes some nice points. Writing is clear. Figure 1 is a really great breakdown of the relative roles of the different forms of disturbance. Figure 2 very nice and well mapped to figure 1.

We thank the reviewer for the overview statements and for the detailed comments that follow.

I have two high-level comments:

First: The authors have a generic “colony specific” form of disturbance, which they argue represents bleaching, thermal stress, coral disease, predators, maybe others. But I suspect this was not actually intended to represent those processes, but was rather conceived of as a generic form of disturbance for comparison with the hydrodynamic disturbances which were (I suspect) the original focus of the modeling. Reading between the lines, it seems that the authors set out to explore the role of hydrodynamic disturbance. They do a nice job of structuring the model to capture some key elements of this disturbance in terms of how it affects different colony sizes differently and how this scaling differs as a function of morphology. And then for comparison they have also developed a second form of disturbance which does not depend on morphology or taxa, and has impacts which are often sub-lethal. This second form of disturbance seems to have been intended as a catchall comparison rather than a detailed representation of a particular disturbance.

The complication for the authors is that they have found that the second, more generic disturbance is actually much more important to community structure than the hydrodynamic disturbance which was the actual focus of their research plan. This creates a challenge for

storytelling because they have to focus on the effects of the generic disturbance but it is not really well mapped on a particular real process.

Unfortunately, this storytelling challenge has tempted the authors into pushing the idea that this generic form of disturbance is really the correct way to represent effects from...well they kind of imply all non-hydrodynamic disturbances... But they list bleaching, thermal stress, coral disease, predators. There are some arguments in the discussion for example around line 408 that such non-physical disturbances have not been shown to have consistently disparate effects on different taxa. But this seems a pretty weak. Bleaching events often affect only a subset of taxa on a reef, with effects that are consequently concentrated on certain morphotypes. Perhaps there are no globally reliable relationships, but I am a little suspicious that the lack of such relationships is a consequence of their lumping so many disturbances together. If they initiated a modeling study looking only a crown of thorns starfish, I expect they would have constructed a model of that disturbance which differs in how it varies with size and morphotype than if they constructed a model of bleaching disturbance. I am not arguing that they redo the modeling and add more disturbances, only that they acknowledge that there is probably more to learn from more focused models of the disturbances that they lump together here (and that their results point towards that as important next steps for such work). I note a few specific places where they ought to dial back their claims below.

We appreciate the commentary of the reviewer. These introductory remarks were quite valuable for us to understand the perspective, and we have re-visited significant sections of our manuscript to assure that we contextualize faithfully. As noted by the reviewer, there are more specific suggestions of locations in the manuscript where such edits would be impactful, and we respond individually to these below.

Second: I found the abstract and intro hard to follow. Did not really understand what was being modeled until the methods. I expect that is partly because of the storytelling complexity discussed above. Lines 22-23 should more clearly state what is being modeled (not environmental forcing which could mean anything). And just in general they should accept their main results and state them more directly:

- hydrodynamic disturbance surprisingly unimportant for community structure
- other disturbance regimes potentially very important; more work needed on those sources

Note that we have significantly edited the Abstract to add emphasis on the findings from hydrodynamical disturbances, which is sage advice. [*Abstract has been largely re-crafted*]

Minor points:

Line 27: Acknowledge that this is not really a model of coralivory, disease or stress. 24-26 do a nice job of stating negative hydrodynamic results, line 27 should just say that more generic disturbances can be strong drivers of morphotype composition. (I am not clear that you showed that “diffusive” or “localized” are the key elements of this generic disturbance, as many factors are different from the hydrodynamic disturbance).

We have re-written the Abstract with this focus on point. We agree that the emphasis on the lack of linkage to hydrodynamic disturbance is the most pointed conclusion, and have rewritten accordingly.

Line 43: where? This sentence seems oddly elliptical.

Understood. We have edited as follows: “*Long-term observations of coral reef benthic community composition offer insights into potential patterns of change under shifting environmental conditions. A 30-year investigation of one well-studied region of the Great Barrier Reef linked the types and scales of disturbance in coral-reef habitats to the structure of the reef benthic community. This work revealed a spectrum of realized community transitions, including limited change, change of coral taxonomic composition, and, in more dramatic cases, competitive shifts of benthic groups from corals to algae or non-coral invertebrates [10].*” [Lines 48-54]

Line 88-101: This model is really cool and nicely described.

Thank you.

Line 102-116: also well described, but I have a little trouble what makes these “colony specific”. The hydrodynamic disturbance is also “colony specific” in that you flip a coin for each colony with weighting based on disturbance intensity and colony size/morphotype? I get the sense that the two disturbance types are different in terms of how correlated the disturbances are across the landscape... can you show that somehow? Perhaps in another figure?

We have introduced text to clarify in text. We appreciate that while the model description does offer the mathematical specifics of the differences among the forms of mortality, a more accessible format holds promise to guide the reader more smoothly into the structure of the model. [Lines 117-122]

Line 124: Growth is proportional to free space in the model. Is this a good approximation given the geometry of real corals? Does (total space – all coral) really map linearly on (opportunities for growth)? I know the Sandin lab has pushed as far as anyone on these kinds of questions so defer to their judgement, but perhaps a sentence addressing this would be appropriate

Indeed, this is an assumption that ignores important elements of spatially explicit growth dynamics. We have tested the robustness of this assumption for a subset of parameter values, creating a spatially explicit model mimicking conditions of the mean-field model presented here. The results are quite similar across a range of parameter values, giving us confidence that the spatial explicitness (on a homogeneous landscape) does not lead to qualitatively different results. We have included a brief description of these findings in the text. [Lines 147-152]

Line 159-160: As discussed above, I am not convinced of this statement for the real world. Perhaps say “ In this model all episodic disturbances...”

Again, this point is well-taken and the proposed edit has been made. [Line 181]

Line 204-205: Fine to say this is beyond the scope, but I would not expect to have alternate states in this model (I do not see any relevant positive feedbacks). You introduce the idea that there might be alternative attractors. Do you see evidence of this? If not might just say so here.

As advised, we have noted that there is no evidence of multiple attractors for any of the models. [Lines 228-229]

Line 208: redundant but not consistent with 200.

This paragraph has been trimmed significantly, thus avoiding redundancy. [Lines 231-237]

Line 224-236: super cool approach. But the tree is just based on means not variance right? That is clear here but gets confusing later.

We have modified the topic sentence of this section regarding the analysis of variance patterns, which we hope will clarify significantly. [Lines 252-254]

Figure 1 is magnificent. But the embedded icons are a little confusing. Could they be placed to show which side has more wave disturbance? As it is they are inconsistent as to which side they are on.

That is a great catch. We have modified the placement of the icons as suggested.

Line 329-330: Is the partial-mortality the key part here?

Your comments have helped us to frame the distinction more cleanly. In fact, the distinction is both the whole- vs partial-to-whole colony mortality and the deterministic vs stochastic pattern of mortality. Hydrodynamic disturbances of a particular magnitude will remove all physically vulnerable colonies (defined by size, morphology, and energy from the disturbance event) during the time interval. Colony-specific mortality is a ‘coin-flip’ to affect a particular colony. We have clarified here and elsewhere in the manuscript. [Lines 117-123 & 344-348]

Line 349-352: this is quite a nice explanation.

Thank you. We really appreciate the positive feedback for particular sections of the manuscript, helping us to learn from ‘successes’ in communication. This is really valuable.

Line 355-356: Just seems too strong a statement. A bunch of assumptions were made here about how the colony specific disturbances scale with colony size (and do not scale with morphology). Assumptions are perfectly reasonable choices but more focus on an individual disturbance (i.e. bleaching) might find literature supporting different choices.

Following the feedback here, we have removed reference to potential biological mechanisms of colony-specific mortality. By focusing this paragraph more squarely on the generic type of ‘colony-specific mortality’, as defined in this manuscript, the paragraph is faithful to the data outputs. We address the caveats of interpretation later in the Discussion. [Lines 371-378]

Line 369-376: super interesting stuff but hard to get this from the figures. Current mapping is very good but does not let one compare assemblage 6 vs 12 (for example). Could each assemblage be mapped on its own (partial) version of figure 2 . Like an 18 panel supplement? Or maybe something cleverer? In any case I think maybe this part could be introduced in the results?

We appreciate the commentary here, yet we worry that this paragraph includes far too much interpretation to be suitable for the Results section. We crafted this paragraph to be an integrated interpretation of the two figures, and as such the content depends upon merging reference to Results with interpretations of meaning. Further, we appreciate the request for an extended figure, but we have not been able to find an approach to re-cast the compositional data without leading to significant redundancy among the figures.

Line 394-412: The fact that local patterns of morphologically specific loss do not hold up in global analyses does not mean these disturbances are not morphologically biased at local scales. I do not find this a convincing argument that the particular choices made in the “colony-specific” disturbances modeled here are more valid than other choices.

We have softened the conclusions of this paragraph, citing that while we may not have globally general ‘rules’ for morphological patterning to colony-specific disturbance, there is potential in localized applications to create a more precise model. Such models would hold promise to provide guidance or predictions of likely futures of reef state. [Lines 427-433]

Reviewer: 3

Comments to the Author(s)

This is a useful paper that contrasts the impacts of system-wide physical disturbance upon corals from 'biological' processes that vary considerably among taxa and with size. A novel aspect of the paper is the categorisation of emergent community state using a clustering approach to link disturbance regime to eventual outcome.

Thank you for the feedback. We are interested in sharing the biological conclusions along with the statistical framework.

The rationale of the study is described clearly. What I found less clear was the parameterisation of non-physical biological disturbances. I agree that it makes sense to generate a broad parameter space of partial and whole-colony mortality and examine how these influence ecological outcomes but I was left wondering whether some of these scenarios are more likely than others. The description of model parameters does not reveal which (if any) of the biological stresses are based on empirical data. Equally, it would be helpful to consider whether some of the outcomes are more representative of specific disturbance scenarios.

We thank the reviewer for the commentary. The comments here map onto the specific commentary of the other reviewers, especially of Reviewer #2. We have sought to soften the interpretation of the disturbance types, in particular in emphasizing the results with reference to hydrodynamic disturbances. The effects of colony-specific disturbances are more nuanced and likely to be case-specific. We encourage the reader to consider such types of disturbances (and potential case-based mapping onto morphological characteristics) through targeted modeling of based on a given geography and coral taxonomic composition. [*Abstract has been re-crafted; Lines 427-433*]

Some differences among taxa can be obtained from Ortiz et al 2014 (Nature Climate Change).

Thank you for pointing us to this paper. We were aware of the findings, and the supplemental information is valuable. We were able to cross-reference many of our parameter estimates with those summarized in this manuscript, and we found strong consensus.

But overall this is a useful contribution to the disturbance literature.

Thank you for the summary review. We are eager to share with the broader community.

Appendix C

UNIVERSITY OF CALIFORNIA, SAN DIEGO

UCSD

BERKELEY • DAVIS • IRVINE • LOS ANGELES • MERCED • RIVERSIDE • SAN DIEGO • SAN FRANCISCO

SANTA BARBARA • SANTA CRUZ

SCRIPPS INSTITUTION OF OCEANOGRAPHY, 0202
CENTER FOR MARINE BIODIVERSITY AND CONSERVATION
TEL: (858) 822- 2790
FAX: (858) 822- 1267

9500 GILMAN DRIVE
LA JOLLA, CALIFORNIA 92093- 0202
EMAIL: CMBC@UCSD.EDU
URL: CMBC.UCSD.EDU

16 September 2020

Dear *RSOS* Editorial Team,

Thank you for reviewing the revised manuscript, "*Modelling the linkage between coral assemblage structure and pattern of environmental forcing*" (Ms. No. RSOS-200565). We appreciate the positive response to the revised manuscript and are pleased to submit this final revised manuscript for consideration by *Royal Society Open Science*.

We are very pleased to read the encouraging comments and support from the three reviewers. Their commentary provided thoughtful insights that led to the original revision. Upon review of the revision, one of the three reviewers asked that we address one outstanding comment from the reviewer's original comments. We apologize for misinterpreting the original review and failing to address the reviewer's concern. Below you will find a detailed response, and we are submitting a revised version of the manuscript that clarifies the text in response. As always, such commentary helps us to provide a clearer and (hopefully) more impactful manuscript.

Thank you again for offering us an opportunity to submit this revised manuscript. We are very encouraged by the reviewers' enthusiasm and hope to share this work more broadly with the academic community soon.

Sincerely,

Stuart Sandin

Professor, Scripps Institution of Oceanography, UC San Diego

Cc: Yoan Eynaud, Gareth Williams, Clinton Edwards, and Dylan McNamara

Comments to authors from reviewers (with responses, **in bold**)

Reviewer: 1

I'm fully satisfied with the authors' response and edits to the MS and congratulate them with this excellent piece of work!

We appreciate the support and enthusiasm of the reviewer. The reviewer's earlier comments contributed greatly to the improved product. Thank you.

Reviewer: 2

This revision is really well done. Paper is great and ready for publication.

The adjustments to the abstract and introduction have really clarified the main points about disturbance. I they retain their arguments on pages 4 and 6 about why the “generic disturbance” is a good representation of disturbances cause by bleaching disease etc, but these make more sense and feel appropriate in the context of the revised framing. It also helps that there is more clarity of the importance of partial vs whole colony mortality, which was in the previous draft but not as cleanly stated.

I accept the author's arguments why the discussion of variance starting around lines 395 needs to go into the discussion. And I take their point that another figure would be hard to add without a lot of redundancy. I just wish it was easier for the reader to connect the stereotypical assemblages from figure 1 to the associated variance results on figure 2. I note the reader can use the regression tree to figure out where each assemblage is and then scribble them in, and my attempt to do that was....well I see how it is hard to visualize :)

Many thanks to the reviewer for the critical thought and the investment in this research. A real challenge for the field is to capture the diversity of community states in transparent and accessible formats. We continue to look for means to simplify the presentation. Following from the reviewer's thinking, it is simply a high-dimensional system that requires some investment to dig into the spectrum of possibility. We thank the reviewer for the dialog and for the support.

Reviewer: 3

I appreciate the efforts the authors have gone to in revising their paper. I think it's almost publishable but my comment 'are some areas of parameter space more likely than others' doesn't seem to have been addressed. I have read the responses to other referees but am still left wondering which areas of parameter space are more realistic or indicative of particular scenarios in situ. The response argues that the language has been softened and that readers are directed to consider more nuanced studies of colony morphology and that these should be modelled specifically in future. Yet the abstract makes strong conclusions about the lack of influence of

hydrodynamic conditions. To me this doesn't reconcile. If you're going to make strong conclusions then I think you need to reflect on the degree to which your model will reflect real-life dynamics.

Our apologies for not attending to this comment in the original revision. Upon re-reading our response to reviewers from the last letter, we recognize that we did not appreciate the intent of the original comment. We now understand that the comment suggested that we provide information on the regions of parameter space that are more or less common.

We have addressed this in two ways. First off, we have defined the range of parameter values that describe the environmental conditions based upon realistic values. In Table 1, we offer reference for the DMT_H range of values, and define the extreme bounds of the other values (e.g., with frequency values ranging from once annually through to effectively never [1 to 0, respectively]). We offer these ranges to define what is possible and realistic for real reefs and have added some text to that effect in the Methods (Lines 213-215 in R2).

Second, we submit that the diversity of environmental conditions across which coral reefs exist does not offer the opportunity to define parameter combinations that are “more likely” than others. We know that there is extreme variation in environments, for example from exposed reef crests to protected lagoons and from corallivory-heavy reefs to benign coral gardens. Based on our observations and our reading of the literature, the forcing environments of each reef need to be considered somewhat individually; there is no ‘normal’. We designed this study understanding that there is a spectrum of conditions today, and individual locations may see changes in these conditions moving forward due to global change. As such, we offered this wide range of environmental conditions for the reader to map their reef onto, and to consider how changes (either across geographic gradients or due to changes through time) could be expected to alter expected coral configurations. We have emphasized this point in the Discussion (Lines 341-345), namely that the broad diversity of environmental conditions defines the broader coral reef ecosystem. We hope that this inspires the reader to understand what is unique about their own reefs, and to consider further how their set of environmental conditions may affect the coral composition.

We thank the reviewer for highlighting this comment, which we apologize for missing earlier. I appreciate the interest in defining the region that is ‘more typical’, but we feel that there is no ‘typical’ reef at all. Instead, we defined the range of environmental conditions to explore what is realistic across the wide variety of coral reefs.

I still believe that the model is useful and appreciate the results - I'd just like to see more thought placed into what is really being represented here.

Thank you for the encouragement and for the attention to detail. The responses to the reviewer's comments have certainly made the manuscript stronger.